# Realistic Synthetic Financial Transactions for Anti-Money Laundering Models

**Erik Altman**[1]   **Jovan Blanuša**[2]   **Luc von Niederhäusern**[2]
**Béni Egressy**[3]*   **Andreea Anghel**[2]   **Kubilay Atasu**[2]
[1]IBM Watson Research, Yorktown Heights, NY, USA
[2]IBM Research Europe, Zurich, Switzerland   [3]ETH Zurich, Switzerland
ealtman@us.ibm.com {jov, lvn}@zurich.ibm.com
begressy@ethz.ch {aan, kat}@zurich.ibm.com

## Abstract

With the widespread digitization of finance and the increasing popularity of cryptocurrencies, the sophistication of fraud schemes devised by cybercriminals is growing. Money laundering – the movement of illicit funds to conceal their origins – can cross bank and national boundaries, producing complex transaction patterns. The UN estimates 2-5% of global GDP or \$0.8 - \$2.0 trillion dollars are laundered globally each year. Unfortunately, real data to train machine learning models to detect laundering is generally not available, and previous synthetic data generators have had significant shortcomings. A realistic, standardized, publicly-available benchmark is needed for comparing models and for the advancement of the area.

To this end, this paper contributes a synthetic financial transaction dataset generator and a set of synthetically generated AML (Anti-Money Laundering) datasets. We have calibrated this agent-based generator to match real transactions as closely as possible and made the datasets public. We describe the generator in detail and demonstrate how the datasets generated can help compare different machine learning models in terms of their AML abilities. In a key way, using synthetic data in these comparisons can be even better than using real data: the ground truth labels are complete, whilst many laundering transactions in real data are never detected.

## 1   Introduction

Money laundering refers to the movement of illicit funds to conceal their origin and make them appear to come from legitimate sources. Laundering schemes often cross bank and national boundaries, producing complex transaction patterns. Even though the amount of laundering is difficult to estimate precisely, the UN puts the global figure at up to \$2 trillion per year [6] as noted in the abstract.

Other, narrower, figures help illustrate the amount of laundering. The CEO of Danske Bank resigned in 2019 after it was revealed that the Estonian branch alone may have laundered \$230 billion [1]. Finextra estimates that in 2017 there may have been \$200 billion in laundering solely in U.S. online sales [4]. Bank fines can be in the billions of dollars for failing to follow stringent standards on laundering [2]. As such, it is valuable to have high-quality data to help build models to detect laundering. Alas, for legal and privacy reasons, real data is not generally available. Even if real data were available, its labeling is inherently poor, as most laundering often goes undetected [7, 5].

**Synthetic financial data** is an emerging area to help address a multitude of challenges with real data and the algorithms used for analysis. These challenges include privacy and differential privacy, competitive advantage, small sample sizes, what-if scenarios, poorly labeled real data, bias, consistent

---

*This work was performed when Béni Egressy was with IBM Research Europe.

37th Conference on Neural Information Processing Systems (NeurIPS 2023) Track on Datasets and Benchmarks.

| Trans. ID | Timestamp | Source bank ID | Source Account | Target bank ID | Target Account | Amount | Currency | Payment type |
|---|---|---|---|---|---|---|---|---|
| 0 | 3 MAY 2019 12:45 | 1 | A | 1 | C | 1400 | USD | Cheque |
| 1 | 15 MAY 2019 07:34 | 2 | B | 1 | C | 710 | EUR | ACH |
| 2 | 18 MAY 2019 16:55 | 3 | E | 1 | C | 950 | USD | Credit card |
| 3 | 1 JUN 2019 10:06 | 1 | C | 3 | D | 1200 | CHF | Wire |
| 4 | 27 JUN 2019 13:18 | 3 | E | 3 | D | 2300 | EUR | Credit card |
| 5 | 7 JUL 2019 11:14 | 3 | D | 1 | A | 1100 | USD | Credit card |
| 6 | 14 JUL 2019 09:37 | 2 | B | 3 | E | 650 | USD | ACH |
| 7 | 20 JUL 2019 14:02 | 3 | E | 3 | D | 2500 | USD | Wire |

(a)

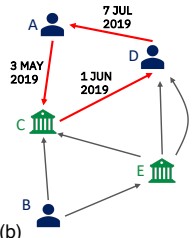

(b)

Figure 1: Financial transactions in (a) tabular format and in (b) graph format.

baseline data for regulators, and more. Realistic synthetic data can help address many of these issues but has its own challenges. For example, does synthetic data match real data and desired scenarios? Can synthetic data be constructed without access to instances of real data being imitated? We address these questions in Sections 2 and 3.

**Relational tables** are the most common way of storing financial data. Figure 1a illustrates financial transactions stored in a tabular format, where each row is a transaction that transfers financial assets from a source account to a destination account, potentially held in different banks. The amount, the timestamp, the payment currency, and the payment type are also indicated for each transaction.

**Graphs** offer a natural representation for financial transaction data. A graph-based data representation exposes the connectivity of the underlying data objects and makes it possible to extract complex patterns of interactions between them. The graph-based representation of transactions shown in Figure 1a is illustrated in Figure 1b. In a financial transaction graph, the nodes typically represent accounts and the directed edges between the accounts represent financial transactions. Several different transactions between the same two accounts can take place at different points in time as illustrated in Figure 1b. Therefore, financial transaction graphs are essentially *directed multigraphs* [10]. This general graph structure underlies many financial crime analysis techniques [63, 43].

**Financial crime** refers to illegal acts committed to obtain financial gain. Financial crime is often linked with suspicious account activities. For instance, in financial transaction graphs, a cycle represents a series of transactions in which the money initially sent from one bank account returns back to the same account; the existence of such cycles is a strong indicator of money laundering [58]. Figure 1b provides a clear visual depiction of cycles in money laundering, which consists of the highlighted transactions from Figure 1a. Similarly, in cryptocurrency transaction networks, criminals use sophisticated mixing and shuffling schemes to cover the traces of their activities [34]. Such schemes can usually be expressed in terms of subgraph structures, such as scatter-gather and bipartite patterns [58, 55, 33, 46] Efficient discovery of such suspicious patterns of interactions across account holders can enable timely and cost-effective detection of criminal activities and their perpetrators.

**Our contributions** can be listed as follows:

- A synthetic financial transaction generator, which builds a multi-agent virtual world in which some of the agents are criminals with illicit income to launder. This synthetic approach provides perfect information about which transactions are laundering and the specific patterns, such as *cycles*, that are used in each instance. This perfect information is not available in real data. Section 3 provides additional details.
- A set of realistic, standardized AML datasets that can be used for developing and benchmarking new money laundering detection models. The datasets cover a range of sizes and difficulties and are all made publicly available.
- A set of initial experiments with Graph Neural Networks (GNNs) and with Gradient Boosted Trees (GBT) that demonstrate the use of the datasets and provide baselines for future work. Importantly, our GNN code is open source. In addition, our GBT results can be reproduced using publicly available tools. Our experimental evaluation is presented in Section 4.
- Observations in Section 7 about how our data release is ethically sound and supports the fight against money laundering and other forms of financial crime.

## 2   Related Work

Despite the importance of good data for money laundering detection, legal, privacy, and competition concerns mean that real financial transaction data is not generally available [52, 31]. Even in the

Table 1: Comparison to key previous synthetic AML data.

| | MLDP | AMLsim | AMLworld (This Work) |
|---|---|---|---|
| Do placement? | ✓ | | ✓ |
| Do layering? | ✓ | ✓ | ✓ |
| Do integration? | ✓ | | ✓ |
| Model 8 key patterns? | | ✓ | ✓ |
| Model other patterns? | ✓ | | ✓ |
| Model multiple banks? | | | ✓ |
| Model multiple currencies? | | | ✓ |
| Complex entity graph? | | | ✓ |
| Model transfers, payments, credits, etc.? | Transfers only | Transfers only | ✓ |
| # of Transactions | 2,340 | 1.32 M | 180 M |
| Laundering rates | 3/5 | 1/762 | 1/807 & 1/1,750 |

restricted cases when anonymized data is available, ground truth labels are not known and data labeling is inherently poor since laundering often goes undetected [7, 5, 6].

For example, one of the few studies to use "real" data is from Starnini et al. [56] which reports 6 months of data from a major Italian bank with 180M transactions. Although myriad laundering patterns are possible, this work looks for only two patterns (aka *smurfs* or *motifs*), and there is no guarantee that all instances of these two patterns are detected. In addition, this data was made available only to Starnini et al. and appears not to be available for use by other investigators.

On a smaller scale, Harris et al. [24] received 2,827 transaction history summaries for 4,469 customers of Scotia Bank in Canada, with the goal of assessing which customers pose a high risk for laundering.

To get around these data availability issues, there have been proposals across many domains to generate synthetic data [53, 45, 44, 32, 40, 61, 37]. However, to the best of our knowledge, existing AML efforts suffer from one or more of the following shortcomings: *(i)* the synthetic data generators need access to real data, which they can then mimic; *(ii)* the AML data lacks authoritative *is-laundering* labels; or *(iii)* the data produced lacks important attributes of real data.

The small set of previous synthetic AML efforts are agent-based, like ours. However, agent-based simulations can vary significantly in their approach, e.g. modeling more or less detail, or using a larger or smaller number of agents. For activities like money laundering, with only a tiny fraction of illicit transactions (i.e. minority class transactions), modeling fidelity and scale matters. Table 1 outlines key aspects of money laundering data. Table 1 also notes the capabilities and differences between our approach *AMLworld*, detailed in Section 3, and two of the most prominent previous efforts to generate synthetic laundering data: *Money Laundering Data Production (MLDP)* [38, 39] and *AMLsim* [63, 57, 9]. As Table 1 suggests, the capabilities of *AMLworld* are significantly more advanced than those of its predecessors, making for more robust data sets. (Terms like *placement*, *layering*, and *integration* will be defined in Section 3, as will the "8 key patterns.") Higher numbers of transactions and lower laundering rates are generally more realistic, and therefore more useful. The 2,340 transactions and 60% laundering rate for *MLDP* suggest a tiny, unrealistic scenario. Section 3.3 provides additional detail about *AMLworld* and *AMLSim* and about the techniques used in *AMLworld* to enhance realism.

GNNs [65, 59, 14, 30, 23, 15, 35, 62] are powerful machine learning models specifically designed for relational data (graphs). In particular, they can also be used for financial crime detection in transaction networks. Cardoso et al. [15] and Weber et al. [64] use GNNs for money laundering detection, Kanezashi et al. [28] use GNNs for phishing detection on the Ethereum blockchain, and Rao et al. [47] uses a GNN to detect fraudulent transactions. Graph Substructure Network, proposed by Bouritsas et al. [14], takes advantage of pre-calculated subgraph pattern counts to improve the expressivity of GNNs. GNNs could also be used to count subgraph patterns, such as in Chen et al. [17], which can enable detecting patterns associated with financial crime.

# 3 Synthetic Datasets

Abundant labeled data is available for key domains such as images, speech, natural language processing, and recommendation engines. However, due to concerns regarding privacy, law, and competitive

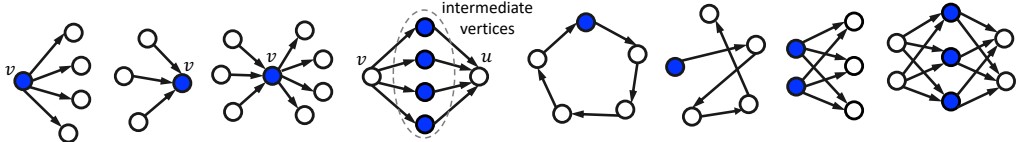

Figure 2: Laundering Patterns Modelled

advantage, little or no financial transaction data is broadly available. To this end, we: (1) introduce our detailed *AMLworld* generator , and (2) make multiple datasets produced by *AMLworld* public.

### 3.1 Motivation for Synthetic Data

Aside from the difficulty of accessing real financial data with laundering transactions, real data has shortcomings that synthetic data can help overcome:

- Individual banks see only their own transactions – not the transactions at the multitude of other institutions with which the bank's customers transact.
- Ground truth labels are usually incomplete, with many instances of laundering being missed. As a result, many AML models mislabel laundering transactions (false negatives).
- Identifying complex money laundering patterns in real data requires additional work – if it is even possible for individual banks (please see the first bullet point).

The conclusive ground truth labels in synthetic data enable the precise comparison of different models. Additionally, the multiple banks in synthetic data enable measurement of how much performance would improve if banks could share their data in a federated learning setup. The scenario of an individual bank can easily be simulated by filtering the synthetic data.

### 3.2 AML patterns

In their *AMLSim* generator, Suzumura and Kanezashi [57] introduced a set of 8 patterns [2] often used in money laundering. These 8 patterns are illustrated in Figure 2. Among its many capabilities outlined in Section 3.3, our *AMLworld* also generates these 8 patterns.

To be more precise, a *fan-out* pattern of vertex $v$, shown in Figure 2a, is defined by the outgoing edges of $v$ connecting $v$ to at least $k \geq 2$ different vertices [58]. Analogously, for a *fan-in* pattern, shown in Figure 2b, $v$ is connected to $k \geq 2$ different vertices via incoming edges. A *gather-scatter* pattern combines a fan-in pattern with a fan-out pattern of the same vertex, as illustrated in Figure 2c [55]. A fan-out pattern of a vertex $v$ and a fan-in pattern of a vertex $u$ form a *scatter-gather* pattern if the fan-out and the fan-in patterns connect vertices $v$ and $u$, respectively, to the same set of intermediate vertices [55]. Figure 2d shows an example of a scatter-gather pattern with four intermediate vertices.

A *simple cycle* pattern, shown in Figure 2e, is a sequence of edges that starts and ends with the same vertex and visits other vertices at most once [27, 12]. Figure 1 shows a cycle in the financial transaction graph, in which node A launders money through nodes C and D. A *Random* pattern, depicted in Figure 2f, is similar to the cycle pattern in Figure 2e – except with random funds are *not* returned to their original account. As such, random can be viewed as a random walk among accounts owned or controlled (e.g. through shell companies). A *bipartite* pattern in Figure 2g moves funds from a set of input accounts to a set of output accounts. Finally, a *stack* pattern in Figure 2h extends the bipartite pattern by adding an additional bipartite layer.

In these cases, as with all the laundering patterns, the laundering entity owns or controls all accounts (nodes) used. For example, an entity may "control" shell companies, that appear to "own" an account. Over the years, rules have become stricter about who is the *Ultimate Beneficial Owner* (UBO) of an account. However, accurate UBO designations can be difficult to enforce or detect. Note also that some entities own or control more nodes than others, and thus can have larger patterns or more variation in their node selection.

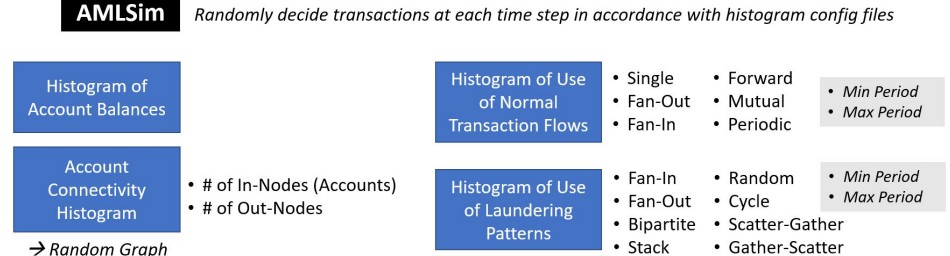

Figure 3: Overview of *AMLSim*.

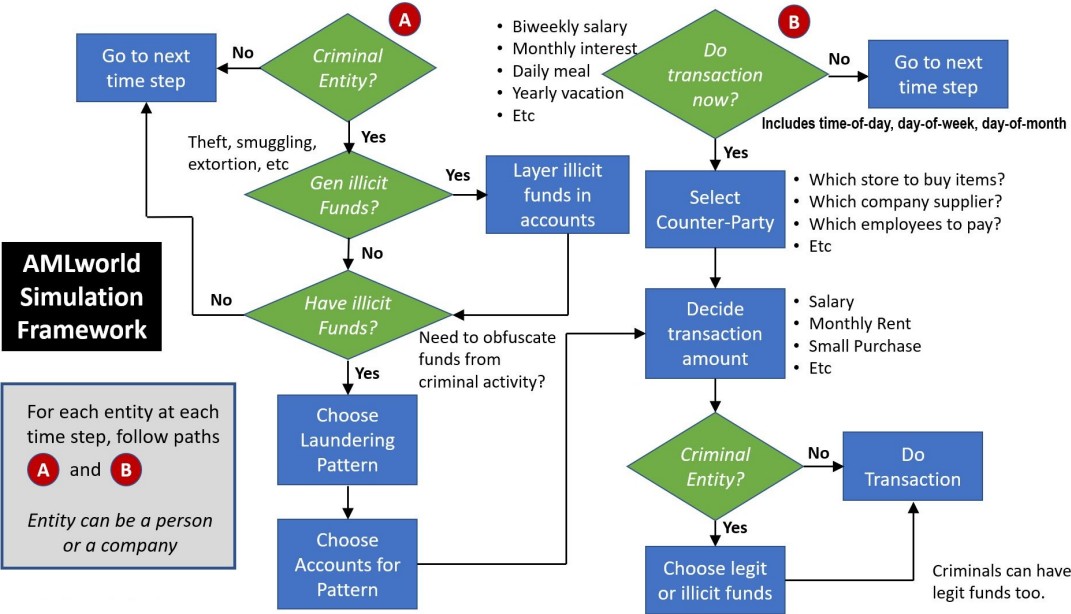

Figure 4: Overview of *AMLworld* Simulation.

## 3.3 What is new in our AMLworld generator?

Our *AMLworld* generator attempts to improve on the pioneering efforts of *AMLSim* [57]. Table 1 in Section 2 gives a breakdown of the differences. For more detail, Figure 3 depicts *AMLSim* and Figure 4 depicts *AMLworld*. As the figures suggest, *AMLworld* models a much richer set of characteristics.

For example, *AMLworld* models the entire money laundering cycle [19]: **(1) Placement**: sources like smuggling of illicit funds; **(2) Layering**: mixing the illicit funds into the financial system; and **(3) Integration**: spending the illicit funds.

*AMLSim* focuses only on (2) Layering – hence the patterns described in Section 3.2. Placement in *AMLworld* can come from 9 sources of criminal activity: extortion, loan sharking, gambling, prostitution, kidnapping, robbery, embezzlement, drugs, and smuggling. Money received and the frequency of these criminal enterprises vary by activity and by the acting entity. Illicit funds are then layered into the financial system (under **A** in Figure 4). Integration happens under **B** at the bottom right of Figure 4 when criminal entities decide how to deploy their funds. *AMLSim* by contrast handles only layering. To support this detailed laundering model *AMLworld* tags illicit funds from their origin (e.g. smuggling proceeds) through all transactions used to launder those funds.

*AMLworld* also has more detailed modeling of population and transaction characteristics, and laundering activity is not limited to specific patterns. Section 3.4 outlines other key attributes of *AMLworld* including the *AMLworld* mechanisms to support "Select Counter Party" at the upper right of Figure 4.

---

[2]Some AML papers refer to patterns as *smurfs* or *motifs*.

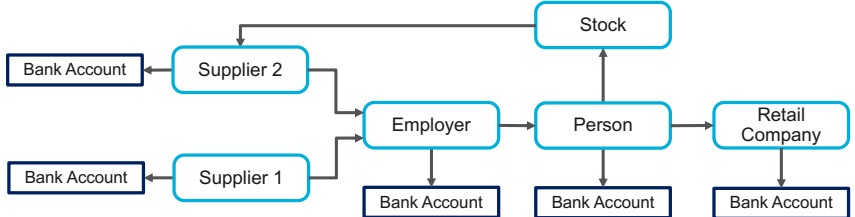

Figure 5: Illustration of the *entity / circular flow graph* representing our Virtual World Model.

## 3.4 The Virtual World Model

At its foundation *AMLworld* builds a multi-agent virtual world of banks, individuals, and companies – with individuals and companies buying items, and making bank transfers to get supplies, pay salaries, pay pensions, etc. The underlying model has good and bad actors, with bad actors doing things like smuggling as noted in Section 3.3.

The bad actors' attempts to launder their illicit funds result in money-laundering transactions. In addition to the laundering patterns discussed above, *AMLworld* also makes some laundering transfers happen "naturally", e.g. a criminal boss launders money by paying employees, buying supplies, etc.

The data produced is a set of transactions as illustrated in Figure 1, with each transaction labeled as laundering (illicit) or not. *AMLworld* has been used to generate billions of transactions in a variety of currencies including dollars, euros, yuan, yen, bitcoin, and more.

The actions of all agents in our virtual world are governed by statistical distributions. Thus the model and data are *not* based on obfuscating or anonymizing real individuals, but are based on real statistics. Likewise, our model uses no seed of real transactions from which to expand. Everything is synthetic. More specifically, as depicted in Figure 5, the underlying virtual world model of *AMLworld* uses a complex entity graph where different entities are connected in many ways. For example, a company serves as an employer, but may also offer retail sales or supply products or services to other companies and individuals. An individual or company may also own other companies or parts (shares) of companies. Each entity (person or company) owns one or more bank accounts directly and via subsidiaries.

The entity graph in Figure 5 is essentially a variant of the *Circular Flow* graphs used to measure GDP (Gross Domestic Product) [3]. As such any financial actor (person or company) can be a (blue) major node, and each such major node can own one or more bank accounts. Edges in the graph reflect financial flows between these nodes. (Flows are often bidirectional, but for simplicity, Figure 5 shows only a subset of important flow directions.)

Transactions are generated based on these relationships and at appropriate frequencies. For example, salaries are typically paid weekly, biweekly, or monthly (as noted in Figure 4). Interest is almost always paid monthly. In addition, a small number of laundering transactions are conducted by entities with funds from illicit activities. Since the generator knows the source of all funds, it knows which transfers are laundering.

Returning to Figure 5, the circular flow graph representing our virtual world illustrates several other important aspects of our approach:

- The circular flow graph enables the generation of a *financial transaction graph* (see Figure 1) – with a list of all transactions between entities during the time period of simulation.
- There are multiple types of entities in the model, including individuals, corporations, partnerships, and sole proprietors. Banks are a special type of entity in the model as they are the locus of most financial transactions among the other types.
- Entities such as people and companies have bank accounts through which their transactions are conducted. Each person and each company may have multiple bank accounts across multiple banks. This spreading can further legitimate pursuits such as saving and spending, but can also facilitate illegal activity such as obfuscating fund sources during laundering.
- Ownership of companies is cross-linked, e.g. company **X** may own company **Y** in whole or part (e.g. through stock). Individuals may also own companies in whole or part. This ownership pattern serves two functions for *AMLworld*: (1) it represents the legitimate complexity of the world, and (2) it allows layers of shell companies to be created through

which bad actors can launder their ill-gotten funds. These shell companies may have accounts at arbitrary banks, e.g. their accounts may be at different banks than the primary bank accounts of their controlling entity.

- The graph structure in Figure 5 is arbitrary, e.g. the graph is not bipartite between merchants and consumers as can be the case in modeling credit card transactions [8].

The *AMLworld* simulator parameterizes population and transaction information to facilitate: (a) tuning; (b) modeling future changes in behavior; and (c) updating to better match real behavior. For (c), authoritative real data is not always available. Feedback loops in *AMLworld* also enforce a degree of tuning to make statistics consistent with observed behavior. For example, if results show too many transactions per person per month, then parameters may be adjusted. Some events are more important than others, e.g. moving money to avoid a negative bank account balance takes precedence in most cases over moving money to get a higher interest rate. Since parameters are *not* all independent, a certain amount of iteration is needed to arrive at a consistent set of parameters.

As suggested by point (b) of the previous paragraph, parameterization is also useful for checking what-if scenarios, e.g. what if there were more laundering, or what if there were changes to banking laws? Synthetic data enables these explorations, which are not possible with real data.

Finally, we note the laundering tag on financial transfers is transitive in *AMLworld*. If **A** pays **B** $100 in illicit funds, and **B** then pays $50 of those funds to **C**, and **C** pays $25 of the $50 to **D**, then not only is the initial $100 considered laundering, but the $50 and the $25 are also. This transitivity is supported by detailed tracking; *AMLworld* propagates this tag through all transactions, including co-mingling of funds from legitimate and non-legitimate sources. *AMLworld*'s complete labeling of laundering is nearly impossible with real-world data.

## 3.5   Public Availability of Synthetic Data

We have used *AMLworld* to create several synthetic AML datasets that have been made available on Kaggle [50]. Data is divided into two high-level groups: **HI** and **LI**, with higher and lower illicit ratios (laundering) respectively. Both **HI** and **LI** are further divided into *small*, *medium*, and *large* datasets, with *large* having $175M - 180M$ transactions. Due to space constraints, Table 4 in the Appendix provides details about the datasets.

However, to demonstrate the realism of *AMLworld*, we discuss several statistics about the **LI**-*Large* dataset. **LI**-*Large* has 176M transactions spanning a little over 3 months. We focus on **LI**-*Large*, but apart from their size and laundering rates, the other five datasets have similar characteristics. Several figures in the Appendix provide grounding about the realism of our data. Figure 7 histograms annualized transactions per account, and its distribution roughly aligns with U.S. Federal Reserve data [51]. Table 5 shows the distribution of transaction formats like ACH, wire, and cheque. Again numbers correspond roughly with Federal Reserve statistics [41].

Overall 1 in 1,750 transactions is laundering in **LI**-*Large*. For comparison, Starnini et al. [56] report on real data with 180M transactions over 6 months from a major Italian bank. They find 855 laundering *motifs* (equivalent to our *Gather-Scatter* and *Scatter-Gather* patterns), each with "less than 20 [nodes] total". This amounts to around 1 laundering transaction per 21,000. By comparison Appendix Table 7 indicates **LI**-*Large* has 8,212 combined *Gather-Scatter* and *Scatter-Gather* transactions – yielding a very similar 1 laundering transaction per 21,440 transactions from these patterns.

## 4   Performance Evaluation of Machine Learning Models

In this section, we provide a preliminary evaluation of the suitability of our synthetic datasets for creating effective machine learning models. A deeper treatment and evaluation is beyond the scope of this *"datasets and benchmarks"* submission, but can be found in the work of Egressy et. al. [20]. In a key way, these measurements are even better than using real data: the ground truth labels are complete, whilst many laundering transactions in real data are never detected.

We train Gradient Boosted Tree (GBT) and popular message-passing Graph Neural Network (GNN) models on the datasets introduced in Section 3.5, using the tabular- and graph-based data representations, respectively. Parameter tuning for GBTs and GNNs is described in Section C of the appendix.

Source and destination account IDs of the transactions are not used as features in our experiments, which prevents our models from recognizing laundering transactions simply by learning account IDs.

**GBT Baselines** We use LightGBM [29] and XGBoost [16] as our GBT baselines, which are widely used machine learning models for tabular data. GBT baseline experiments were performed using version 3.1.1 of LightGBM and version 1.7.6 of XGBoost. To improve the performance of these GBT models, we use Graph Feature Preprocessor (GFP) [48, 49], which creates additional features for the datasets. To achieve this, GFP interprets the input data as a graph and extracts various graph-based features from this graph, such as vertex statistics and number of simple cycles [13, 12]. As a result, GBT models can exploit the underlying graph structure of the datasets. Section D of the appendix provides details on how GFP was configured for the experimental evaluation.

**GNN Baselines** GIN (Graph Isomorphism Network) with edge features [65, 25] and PNA (Principal Neighbourhood Aggregation) [60, 18] are used as GNN baselines. Since AML is a transaction classification problem, we also include a baseline using edge updates (GIN+EU) [11]. This approach is similar to the architecture used by Cardoso et al. [15], which has recently been used in self-supervised money laundering detection. Note that the GNN results for the *Large* datasets are not available because training models for these datasets requires significantly more compute resources.

Because the vertices are accounts and the edges are transactions in our financial transaction graph representation, detection of illicit transactions is an edge classification problem. All our GNNs use a final edge readout layer that performs classification using the edge embedding and the respective endpoint node embeddings as input. To reduce the complexity, we use neighborhood sampling [23] when training and testing our GNN models. We sample 100 one-hop and 100 two-hop neighbors.

**Data Split** We use a 60-20-20 temporal train-validation-test split, i.e., we split the transaction indices after ordering them by their timestamps. The data split is defined by two timestamps, $t_1$ and $t_2$. Train indices correspond to transactions before time $t_1$, validation indices to transactions between times $t_1$ and $t_2$, and test indices to transactions after $t_2$. However, in the GNN case, the validation and test set transactions need access to the previous transactions to identify patterns. So we construct train, validation, and test graphs. This corresponds to considering the financial transaction graph as a dynamic graph and taking three snapshots at times $t_1$, $t_2$, and $t_3 = t_{max}$. The train graph contains only the training transactions (and corresponding nodes). The validation graph contains the training and validation transactions, but only the validation indices are used for evaluation. The test graph contains all the transactions, but only the test indices are used for evaluation. This is the most likely scenario faced by banks and financial authorities when classifying unseen batches of transactions.

**Evaluation of GBT and GNN Baselines** As our proposed datasets are naturally very imbalanced, traditional metrics like accuracy do not provide a reliable measure of model performance. In this context, we have chosen to emphasize the F1 score for the minority class. Detailed precision and recall scores for the GNN baselines can be found in appendix Section E. Table 2 showcases the inference performance for the range of models evaluated. These results show that message-passing GNNs that exploit the connectivity between accounts are effective in capturing laundering transactions. Advanced GNN architectures such as PNA and GIN+EU, which respectively implement extended message aggregation and edge update mechanisms, significantly enhance the GNN performance. Furthermore, the combination of GFP with gradient boosting techniques, specifically LightGBM and XGBoost, also demonstrates strong performance across different datasets. An interesting observation is the near-par performance of PNA with GBT baselines, even without relying on handcrafted features.

The **LI** datasets present more significant predictive challenges compared to the **HI** datasets. This is primarily attributed to their lower illicit ratios as well as the longer time span of their laundering patterns. The primary difference between **LI** and **HI** datasets is that criminals launder less frequently in **LI**, which makes it harder to discover the laundering patterns. To address this, our models incorporate the measure of weighting the predictions of the minority class higher in the loss function. However, considering the intricacies of dealing with the **LI** datasets, exploring other strategies, such as the Pick and Chose method [36], which employs a label-balanced sampler, or GraphSMOTE [66], which introduces synthetic minority class samples in the embedding space, may be imperative.

The prediction accuracy on the more challenging **LI** datasets can be improved by reusing or fine-tuning models pretrained on the *HI* datasets. Table 3 shows that reusing the pretrained LightGBM and PNA models on the *HI-Medium* and *-Large* datasets enables higher F1 scores than those of XGBoost models trained on the respective **LI** datasets (see Table 2). Furthermore, the classification

Table 2: Minority class F1 scores (%). HI indicates a higher illicit ratio. LI indicates a lower ratio.

| Model | **HI**-*Small* | **LI**-*Small* | **HI**-*Medium* | **LI**-*Medium* | **HI**-*Large* | **LI**-*Large* |
|---|---|---|---|---|---|---|
| GIN [65, 25] | $28.70 \pm 1.13$ | $7.90 \pm 2.78$ | $42.30 \pm 0.44$ | $3.86 \pm 3.62$ | NA | NA |
| GIN + EU [11, 15] | $47.73 \pm 7.86$ | $20.62 \pm 2.41$ | $49.26 \pm 4.02$ | $6.19 \pm 8.32$ | NA | NA |
| PNA [60] | $56.77 \pm 2.41$ | $16.45 \pm 1.46$ | $59.71 \pm 1.91$ | $27.73 \pm 1.65$ | NA | NA |
| GFP [48, 49] + LightGBM [29] | $62.86 \pm 0.25$ | $20.83 \pm 1.50$ | $59.48 \pm 0.15$ | $20.85 \pm 0.38$ | $48.67 \pm 0.24$ | $17.09 \pm 0.46$ |
| GFP [48, 49] + XGBoost [16] | $63.23 \pm 0.17$ | $27.30 \pm 0.33$ | $65.70 \pm 0.26$ | $28.16 \pm 0.14$ | $42.68 \pm 12.93$ | $24.23 \pm 0.12$ |

Table 3: Minority class F1 scores (%) of models trained on HI datasets and evaluated on LI datasets.

| Model | **LI**-*Small* | **LI**-*Medium* | **LI**-*Large* |
|---|---|---|---|
| PNA [60] | $0.00 \pm 0.00$ | $36.60 \pm 0.74$ | NA |
| PNA [60] + fine-tuning (5 epochs) | $27.38 \pm 1.03$ | $36.84 \pm 1.64$ | NA |
| GFP [48, 49] + LightGBM [29] | $0.02 \pm 0.03$ | $32.51 \pm 0.24$ | $21.77 \pm 0.15$ |
| GFP [48, 49] + XGBoost [16] | $0.00 \pm 0.00$ | $30.31 \pm 0.28$ | $17.48 \pm 6.90$ |

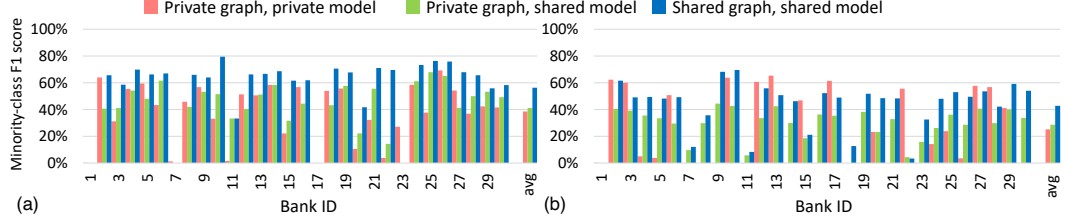

Figure 6: Effect of data and model sharing across banks for (a) **HI**-*Medium* and (b) **HI**-*Large* datasets.

performance on **LI** data can be improved further for the **LI**-*Small* dataset, where directly using the models trained on **HI** datasets is ineffective, by fine-tuning the **HI** PNA models using the **LI** dataset.

In Figure 6, we report results on **HI**-*Medium* and **HI**-*Large* datasets for individual banks using GFP to create features and LightGBM to build models. Related experiments performed on the **LI**-*Medium* and **LI**-*Large* datasets can be found in Section F of the appendix. For brevity, we focus on the top-30 banks that participate in the largest number of transactions in each dataset. We evaluate three setups:

- In the *shared graph, shared model* setup, banks share all their transactions with each other to create a shared financial transaction graph. The GFP is applied to this shared graph to extract graph features, which are also shared. A shared LightGBM model is built using this shared data, which the banks use to score their own transactions. Note that the results given in Table 2 show the aggregate F1 score for all the banks produced using the same setup.
- The *private graph, private model* setup computes the F1 scores using the LightGBM models trained separately by each bank on its private data. In this setup, a bank sees only the transactions performed by its own account holders. A private graph is constructed using these transactions, and the GFP is executed on this private graph to extract features. In addition, each bank trains its own model on its own data, so there is no model sharing either.
- In the *private graph, shared model* setup, each bank creates a private transaction graph and extracts graph features from it using GFP as in the previous case. The banks do not share the source and destination account ids with each other, but share the remaining transaction features, including the graph features. A shared LightGBM model is trained using this data.

Figure 6 shows that, in general, financial institutions can achieve higher F1 scores by building shared machine learning models. However, building shared models on top of shared financial transaction graphs leads to even more significant improvements. To achieve such accuracy improvements in a privacy-preserving manner, financial institutions can use differentially-private model and topology sharing techniques. Even though evaluation of such techniques is outside the scope of our work, we believe that our contributed datasets offer a valuable testing ground for building such approaches.

## 5 Future Work and Research Avenues

The focus of this paper is the generation of high-quality datasets and their use to build effective anti-money laundering models. Here are some additional research opportunities that follow naturally:

1. Historically, AML models have focused on transactions at one bank – because that is the only data available to individual banks that review transactions for laundering. Our synthetic

data enables analysis across multiple banks as shown in Figure 6. With that comes a need to build privacy-preserving machine learning models and graph topology-sharing techniques.

2. Whether considering transactions at one bank or across banks, there is a need for machine learning and deep learning models that can detect complex AML patterns in graph datasets.

3. Building on points (1) and (2), using deep neural networks, it may be possible to use our synthetic cross-bank data for pre-training, before fine-tuning with real-world data. Our initial experiments given in Table 3 demonstrate that transfer learning is a viable approach.

4. As new models and algorithms are developed, it is important that they run efficiently in terms of time, memory consumption, CPU and GPU use, etc. Techniques to achieve such efficiency when analyzing financial transactions differ from those needed in other domains.

5. Others may find ways to improve our data generation method. For example, with our agent-based approach, there can be challenges around the stability of the data generation: minor parameter tweaks can sometimes yield significant changes in aggregate results, such as the laundering rate. Generative models for graphs could be a promising research avenue.

## 6  Conclusions

We have outlined *AMLworld*, a detailed, multi-agent, virtual world approach to generating synthetic financial data labeled for money laundering. We have established that the generated data matches real-world data in key regards. *AMLworld* data also has perfect tagging for laundering activity – something essentially impossible with real data, where tags typically miss a great deal of laundering activity. By providing realistic tagged data, *AMLworld* facilitates the development of money laundering detection models, and our experiments provide baseline scores, demonstrating the utility and efficacy of machine learning models. Lastly, *AMLworld* enables the generation of models to detect *what-if* laundering patterns not yet observed in real data. Six *AMLworld* datasets are now publicly available on Kaggle [50]. Our initial evaluation on these datasets shows that GNNs and GBTs can be effective solutions for identifying laundering transactions. While GBTs require some feature engineering to capture complex laundering patterns, GNNs often produce competitive results without any feature engineering. Further research is needed to enable collaboration and cooperation between financial institutions through the use of differentially-private graph topology and model sharing techniques.

## 7  Ethics

We believe that our release of this synthetic laundering data *helps* in the AML fight.

As noted in the Introduction, most laundering activity currently goes undetected. This suggests the malefactors have the upper hand – as opposed to domains like credit card fraud, where the large majority of fraud is detected and quickly. One key goal in releasing this data is to facilitate improvements in laundering detection to enable results closer to credit card fraud detection.

A key advantage for credit card fraud detection is that a close ground truth is well understood – via customer feedback about unpurchased items. Unfortunately, no good ground truth exists for laundering activities in real financial data – leading to high error rates and much missed fraud. By contrast, our synthetic data has this ground truth that has been so helpful against credit card fraud.

Beyond helping the good side, we believe our data has limited potential to help the bad side. An obvious way we might do that is by publishing the algorithms used by bank X to detect laundering. Launderers could then change their behavior so as to avoid being flagged by the algorithms. With only a static set of data, it is much harder for launderers to change their behavior to avoid detection.

To this point, we emphasize that what we have open-sourced is a set of data – *not* the code to generate the data. If launderers had our source code, it would be easier for them to tweak the data produced and check the results versus whatever detection algorithms they could get their hands on.

We offer a few additional observations on ethics in Appendix B.

## Acknowledgments and Disclosure of Funding

The support of the Swiss National Science Foundation (project numbers: 172610 and 212158) for this work is gratefully acknowledged.

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

# Supplementary Material

## A   Realism of Data Generation

As described in Section 3.5, we used *AMLworld* to create several synthetic AML datasets. The datasets are published on Kaggle [50] under a Community Data License Agreement. As noted previously, data is divided into two high-level groups: **HI** and **LI**. The **HI** datasets have slightly higher illicit ratios (more laundering) than **LI**. Both **HI** and **LI** are further divided into *small, medium,* and *large* datasets, with *large* having $175M - 180M$ transactions. Table 4 provides a number of basic statistics about all six Kaggle datasets. We include more detailed analyses below.

Table 4: Public Synthetic Data Statistics. **HI** = Higher Illicit (more laundering). **LI** = Lower Illicit.

|                                  | Small | | Medium | | Large | |
| --- | :---: | :---: | :---: | :---: | :---: | :---: |
| **Statistic**                    | **HI** | **LI** | **HI** | **LI** | **HI** | **LI** |
| # of Days Spanned                | 10    | 10    | 16    | 16    | 97    | 97    |
| # of Bank Accounts               | 515K  | 705K  | 2077K | 2028K | 2116K | 2064K |
| # of Transactions                | 5M    | 7M    | 32M   | 31M   | 180M  | 176M  |
| # of Laundering Transactions     | 3.6K  | 4.0K  | 35K   | 16K   | 223K  | 100K  |
| Laundering Rate (1 per N Trans)  | 981   | 1942  | 905   | 1948  | 807   | 1750  |

Figure 7 histograms annualized transactions per account. It gives a micro view of how different entities drive the overall transaction counts found in Table 4. Figure 7 also provides a touch point to real data: numbers roughly align with U.S. Federal Reserve data [51]. Table 5 shows the distribution of transaction formats used in the dataset, such as ACH, wire, and cheque. Again numbers correspond roughly with Federal Reserve statistics [41].

Table 8 breaks down laundering transactions between (a) transactions following one of 8 standard patterns discussed in Section 3.2 – in particular Figure 2; and (b) the *integration* transactions disguised as other activities (e.g. employee payroll or company supplies).

As with other data, we base salary and pension amounts on real data, in this case from the US Internal Revenue Service [54]. Figure 8 shows the number of returns filed for the 2018 tax year for each amount of salary income and pension income. Having proper distribution of salaries and pensions in turn helps drive accurate modeling and statistics for transaction sizes and frequency, as just discussed. The $0 top bin in Figure 8 reflects the fact that some people have no salary income or no pension income. Beyond salary and pension people can have income from other sources such as interest or dividends.

Table 5: Distribution of transaction counts by format in **LI**-*Large*.

| Format | # Transactions | Format | # Transactions |
|---|---|---|---|
| Cheque | 69,720,485 | Reinvestment | 7,258,251 |
| Credit Card | 49,923,366 | Wire | 6,346,402 |
| ACH | 21,650,558 | Bitcoin | 3,601,817 |
| Cash | 18,069,965 | | |

Table 6: Histogram of # of nodes (accounts) in **LI**-*Large* laundering patterns. **Gather Scatter** has 2 counts: (a) # of nodes from which initial funds come; (b) # of nodes to which funds ultimately go.

| # of Nodes | | Fan-out | Fan-in | Cycle | Random | Bipartite | Stack | Scatter-Gather | Gather-Scatter | |
|---|---|---|---|---|---|---|---|---|---|---|
| Min | Max | | | | | | | | (1) | (2) |
| 1 | 2 | 50 | 47 | 72 | 70 | 72 | 87 | 54 | 67 | 67 |
| 2 | 4 | 46 | 54 | 50 | 44 | 30 | 31 | 47 | 43 | 38 |
| 4 | 8 | 53 | 54 | 84 | 79 | 62 | 56 | 64 | 61 | 55 |
| 8 | 12 | 57 | 62 | 76 | 62 | 51 | 48 | 48 | 42 | 52 |
| 12 | 18 | 71 | 62 | 16 | 23 | 62 | 37 | 63 | 71 | 72 |
| 18 | ∞ | 0 | 0 | 0 | 0 | 0 | 0 | 0 | 0 | 0 |

Table 7: Occurrence count for each laundering pattern in **LI**-*Large*.

| Pattern | Pattern Count | # Trans in Pattern | Pattern | Pattern Count | # Trans in Pattern |
|---|---|---|---|---|---|
| Fan-out | 277 | 2,014 | Stack | 259 | 3,239 |
| Fan-in | 279 | 2,003 | Random | 278 | 1,831 |
| Cycle | 298 | 2,326 | *Scatter-gather* | 276 | 4,202 |
| Bipartite | 277 | 1,858 | *Gather-scatter* | 284 | 4,010 |

Table 8: Laundering rates in **LI**-*Large*. **Ratio** is the total transaction count divided by the laundering transaction count.

| Description | # of Trans | Ratio |
|---|---|---|
| Laundering Trans. - Patterns | 19,461 | 9,047 |
| Laundering Trans. - Other | 81,143 | 2,170 |
| Laundering Trans. - Total | 100,604 | 1,750 |

Figure 8 indicates that there are about 3.4× as many returns showing salary income as pension income, although returns can show both [3]. We assume about 62.5% of people in our models have salaries – matching the value from the U.S. Department of Labor for labor force participation of the adult workforce [42]. Following the IRS ratios, about 18.3% of the population in our datasets has pension income, with around half of those pensioners also receiving a salary.

[3]For income tax purposes in the U.S., salary income also includes total wages for hourly workers.

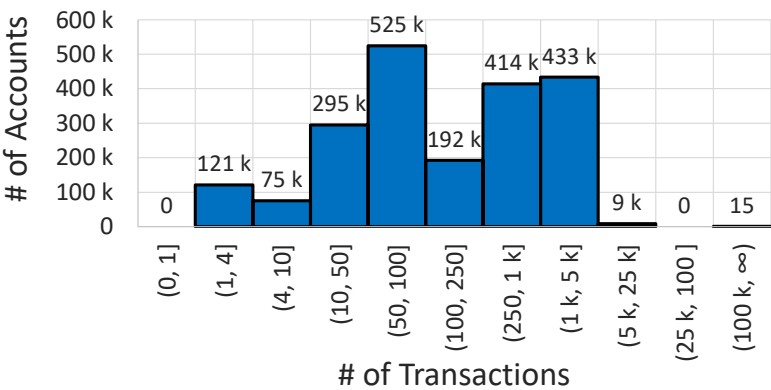

Figure 7: Annualized transaction rate across accounts in **LI**-*Large*.

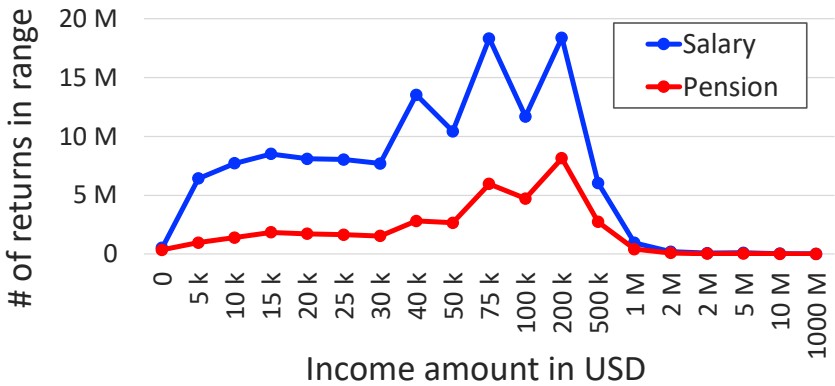

Figure 8: Distribution of U.S. tax returns for 2018 by salary income and pension income.

Table 9: Successive halving parameter configurations used for hyperparameter tuning of shared models trained using all banks of a dataset (*Multi-bank models*) and private models trained using the data of a single bank (*Single-bank models*).

| | Multi-bank models | | | Single-bank models | |
|---|---|---|---|---|---|
| **Datasets** | *Small* | *Medium* | *Large* | *Medium* | *Large* |
| $x_0$ | 1000 | 100 | 16 | 1000 | 100 |
| $\eta$ | 2 | 2 | 2 | 1.5 | 2 |
| $r_0$ | 0.1 | 0.1 | 0.2 | 0.3 | 0.1 |

Additional statistics about our virtual world were provided in Section 3.5.

## B   Ethical Use of the Data

We addressed a number ethics issues in Section 7. We offer a few additional observations here.

Ethical use of the data includes using it for benchmarking and improving models for the detection of money laundering activity. We foresee significant positive societal impacts from such uses of our data. Money laundering has a huge cost for society in itself, but more importantly, money laundering enables a whole range of criminal activities to continue, ranging from phishing attacks to human trafficking. Detecting money laundering transactions can help authorities uncover such activities and identify the criminals behind them. Additionally, the data could be used for pretraining detection models before fine-tuning on real data.

Given that the dataset is synthetic, there is no risk of it containing personally identifiable information or offensive content. Moreover, researchers do not need to take special care with its use, but they should bear in mind that performance might not translate one-to-one to real data.

## C   Hyperparameter Tuning of Machine Learning Models

**GBT Baselines.** We use successive halving [26] for the hyperparameter tuning of the LightGBM and XGBoost models. Successive halving starts by randomly sampling $x_0$ model parameter configurations. Each configuration of model parameters is evaluated using a fraction $r_0 \leq 1$ of the initial training set. The algorithm then finds the best $x_0/\eta$ configurations, with $\eta > 1$. These best configurations are used in the next round of successive halving using a fraction $\eta \times r_0$ of the train set. This process continues until the fraction of the training set used for the evaluation reaches 1. We use different successive halving configurations for different datasets, as shown in Table 9. Furthermore, the parameter ranges from which $x_0$ initial parameter configurations are sampled are shown in Table 10.

**GNN Baselines.** We used random sampling to identify a good range of GNN hyperparameters. A second round of random sampling was conducted with this narrower range to pick our final set of hyperparameters. We varied the following hyperparameters: the number of GNN layers, hidden embedding size, learning rate, dropout, and minority class weight (for the weighted loss function).

Table 10: Model parameter ranges used at for the hyperparameter tuning of GBT models. The small parameter range, indicated with *Range-small*, is only used for hyperparameter tuning of LightGBM models related to **HI**-*Large* and **LI**-*Large* datasets. The large parameter range shown in column *Range-large* is used for other datasets.

| LightGBM | | | XGBoost | |
| --- | --- | --- | --- | --- |
| **Parameter** | **Range-small** | **Range-large** | **Parameter** | **Range** |
| num_round | $(32, 512)$ | $(10, 1000)$ | num_round | $(10, 1000)$ |
| num_leaves | $(32, 256)$ | $(1, 16384)$ | max_depth | $(1, 15)$ |
| learning_rate | $(0.001, 0.01)$ | $10^{(-2.5,-1)}$ | learning_rate | $10^{(-2.5,-1)}$ |
| lambda_l2 | $(0.01, 0.5)$ | $10^{(-2,2)}$ | lambda | $10^{(-2,2)}$ |
| scale_pos_weight | $(1, 10)$ | $(1, 10)$ | scale_pos_weight | $(1, 10)$ |
| lambda_l1 | $10^{(0.01,0.5)}$ | $10^{(0.01,0.5)}$ | colsample_bytree | $(0.5, 1.0)$ |
| | | | subsample | $(0.5, 1.0)$ |

The exact ranges we used are listed in 11. The number of random samples was set to between 10 to 50, depending on the training time of the model on a particular dataset. To get our final results, we use the hyperparameters with the best validation score to train four models initialized with different random seeds.

Table 11: Model parameter ranges used for hyperparameter tuning of GNN models. The hyperparameters were optimized on the small datasets and used for all GNN models.

| **Parameter Ranges** | **GIN (+EU)** | **PNA** |
| --- | --- | --- |
| hidden embedding size | (16,72) | (16,64) |
| learning rate | (0.005,0.05) | (0.0001, 0.002) |
| number of GNN layers | (2,4) | (2,4) |
| dropout | (0,0.5) | (0, 0.2) |
| minority class weight | (6,8) | (6,8) |

# D   Graph Feature Preprocessor Configuration

The Graph Feature Preprocessor (GFP) processes transactions represented as temporal edges in a streaming fashion. The input of this preprocessor is a batch of temporal edges that the preprocessor inserts into its in-memory dynamic graph and extracts various graph features from this graph, such as scatter-gather patterns, simple cycles, and vertex statistics. The output of the preprocessor is the same batch of edges with these additional graph-based features. This library is available as part of the Snap ML library [49], and the experiments in this paper were performed using version 1.14 of Snap ML. More details on GFP are available in the documentation [48].

The graph-based features for our experiments are extracted using the batch size of 128. The GFP is configured to generate features based on scatter-gather patterns, temporal cycles, simple cycles of length up to 10, and vertex statistics. We set the GFP to use a time window of six hours for scatter-gather patterns and a time window of one day for the rest of the graph-based features. The vertex statistic features are computed using the "Amount" and "Timestamp" fields of the basic transaction features (see Figure 1a). This configuration is used for all datasets and experiments that use GFP in this paper.

For each dataset, GFP processes transactions in the increasing order of their timestamps. This ordering ensures that the graph-based features for each transaction are extracted using the past data. As a result, transactions from the training set will not contain graph-based features computed using the validation or test sets. Processing transactions in such a way prevents data leakage.

# E    Additional GNN Experiments

Our GNN code is included with the supplementary material and available publicly on GitHub [4] under an Apache License. The GNNs are implemented using PyTorch Geometric version 2.3.1 [21] and PyTorch version 2.0.1.

All the baseline GNN experiments were run on an internal cluster on Nvidia Tesla V100 GPUs. Table 12 shows the runtimes when training the different GNN baselines on the AML Small and Medium datasets. The size of the GNN models was kept the same: 2 GNN layers and a hidden embedding size of 64. The total GPU time, including initial experiments and hyperparameter optimization, is estimated to have been around 1000 GPU hours.

Table 12: Total training times (TTT) and inference performance in Transaction per Second (TPS) for all GNN baselines on the AML Small and Medium datasets using an Nvidia Tesla V100 GPU.

| Model | **HI**-*Small* | | **LI**-*Small* | | **HI**-*Medium* | | **LI**-*Medium* | |
|-------|--------|-----|--------|-----|--------|-----|--------|-----|
|       | TTT (s) | TPS | TTT (s) | TPS | TTT (s) | TPS | TTT (s) | TPS |
| GIN | 22703 | 17210 | 29655 | 15784 | 85652 | 7101 | 85994 | 7316 |
| GIN+EU | 26046 | 11844 | 36625 | 11640 | 85753 | 4981 | 85887 | 5097 |
| PNA | 27745 | 16557 | 39648 | 14994 | 85654 | 6725 | 85827 | 6819 |

In addition to the minority class F1 scores in Table 2, we include more fine-grained results detailing the precision and recall rates of the GNN-based models. Precision evaluates the accuracy of laundering predictions, recall measures the model's ability to identify all laundering instances, while the F1 score is the harmonic mean of precision and recall. Table 13 shows the laundering recall rates and Table 14 shows the corresponding precision scores. In Figure 9, we give example precision-recall curves, which visually capture the trade-off between precision and recall across different decision thresholds. The examples are taken from the best-performing seed from the best-performing model (PNA) on all small and medium AML datasets.

Table 13: Minority class recall rate (%) for the GNN-based models. HI indicates a higher illicit ratio. LI indicates a lower illicit ratio.

| Model | **HI**-*Small* | **LI**-*Small* | **HI**-*Medium* | **LI**-*Medium* |
|-------|---------|---------|---------|---------|
| GIN [65, 25] | $38.16 \pm 5.92$ | $14.59 \pm 2.37$ | $39.86 \pm 3.61$ | $8.07 \pm 9.32$ |
| GIN+EU [11, 15] | $55.41 \pm 5.96$ | $23.26 \pm 2.87$ | $48.06 \pm 6.45$ | $5.51 \pm 6.82$ |
| PNA [60] | $53.15 \pm 2.26$ | $16.43 \pm 2.62$ | $47.42 \pm 4.30$ | $20.44 \pm 0.66$ |

Table 14: Minority class precision (%) for the GNN-based models. HI indicates a higher illicit ratio. LI indicates a lower illicit ratio.

| Model | **HI**-*Small* | **LI**-*Small* | **HI**-*Medium* | **LI**-*Medium* |
|-------|---------|---------|---------|---------|
| GIN [65, 25] | $27.40 \pm 7.98$ | $5.14 \pm 3.42$ | $47.78 \pm 5.20$ | $5.23 \pm 3.60$ |
| GIN+EU [11, 15] | $42.29 \pm 10.69$ | $21.60 \pm 10.83$ | $52.56 \pm 8.27$ | $12.13 \pm 19.35$ |
| PNA [60] | $58.48 \pm 10.67$ | $17.37 \pm 5.80$ | $69.12 \pm 4.26$ | $36.50 \pm 10.51$ |

# F    Additional GBT Experiments

Figure 11 shows the precision-recall curves for XGBoost models trained on all AML datasets using graph-based features created with Graph Feature Preprocessor. These curves correspond to the results from the row "GFP+XGBoost" shown in Table 2. The steep slope of each curve after the red dot, which represent a data point obtained using the prediction threshold of 0.5, indicates that it is challenging to obtain higher recall without significantly degrading the precision. Note that it is possible to achieve higher precision of these XGBoost models by simply reducing the recall by a few percent.

---

[4]https://github.com/IBM/Multi-GNN

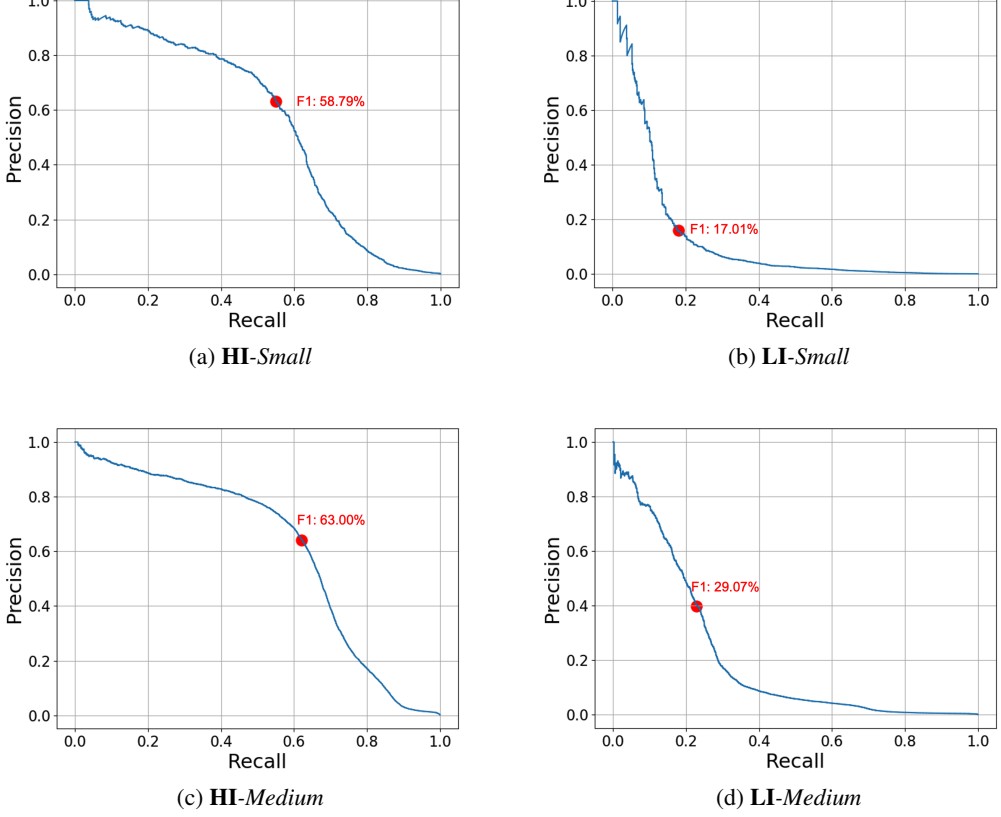

Figure 9: Precision-Recall Curves for the best-performing PNA model for all AML datasets. The red dot indicates the F1 score obtained using the prediction threshold of 0.5.

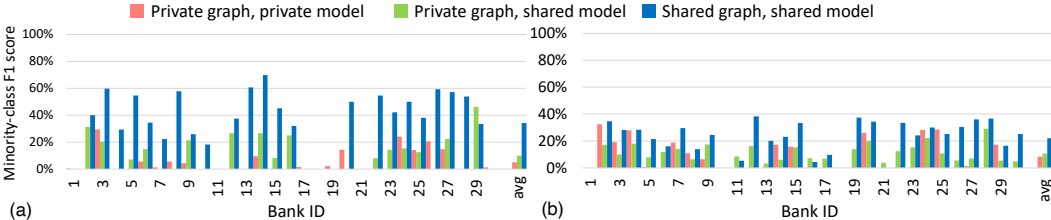

Figure 10: Effect of sharing the data and LightGBM models across banks for the (a) **LI**-*Medium* and (b) **LI**-*Large* datasets.

Figure 10 shows performance on a per-bank basis using the **LI** datasets. The experiment is explained in Section 4 and the plot is analogous to Figure 6, but here we are using the **LI**-*Medium* and **LI**-*Large* datasets. These dataset contain fewer illicit transactions compared to the **HI**-*Medium* and **HI**-*Large* datasets (see Table 4). Therefore, it is more challenging to build a machine learning model for a bank using only local data of these **LI** datasets compared to the banks of **HI** datasets. In this case, the average minority-class F1 score across 30 banks shown in Figure 10 is only 4.9% for **LI**-*Medium* and 8.7% for **LI**-*Large*. Nevertheless, the improvement in minority-class F1 score of the *shared graph, shared model* case compared to the *private graph, private model* case is still significant. Sharing the transaction graph and the global model across the banks increases the average minority-class F1 score to 20.8% for **LI**-*Medium* and to 22.1% for **LI**-*Large*.

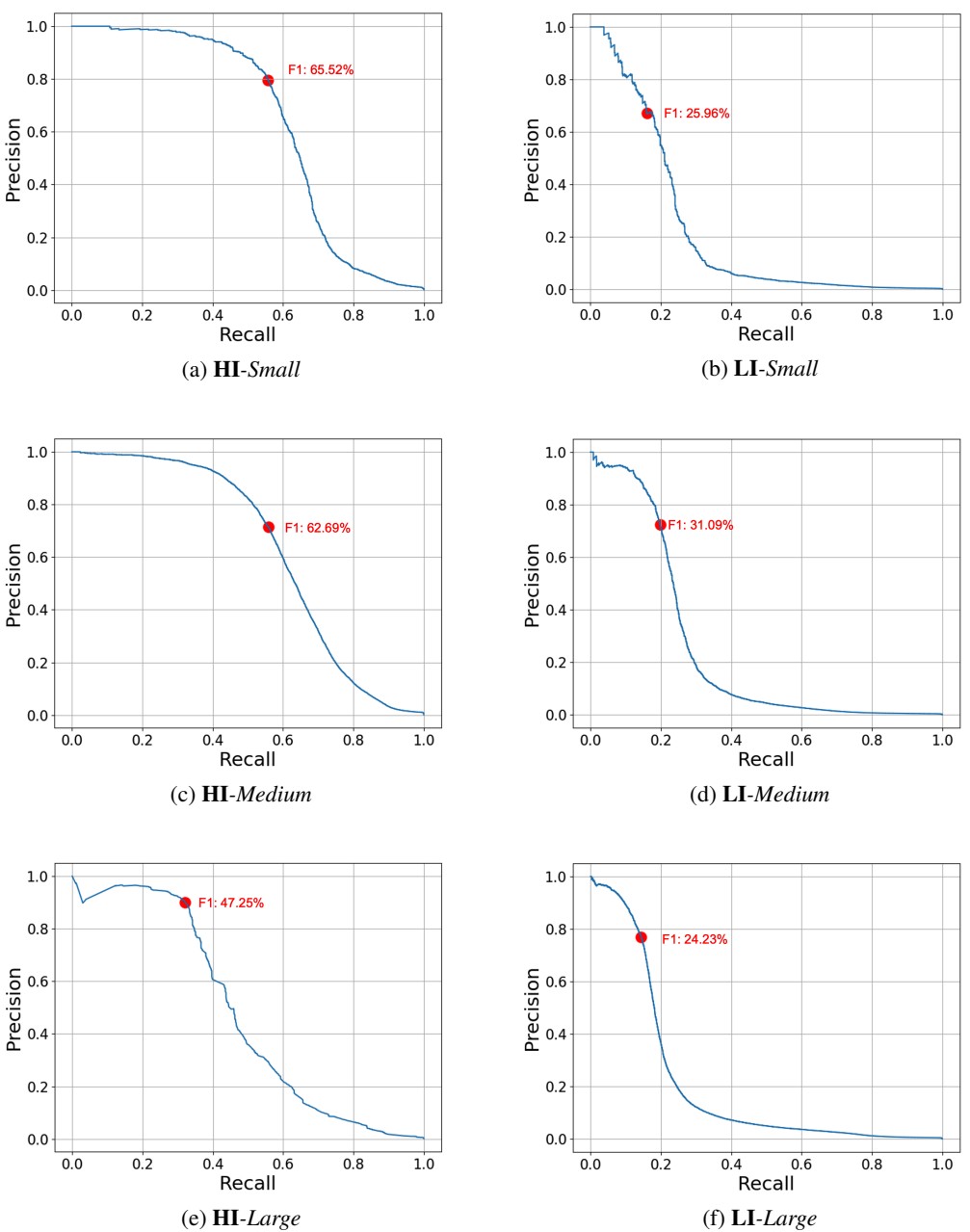

Figure 11: Precision-Recall Curves for the XGBoost models for all AML datasets trained using graph-based features of GFP. The red dot indicates the F1 score obtained using the prediction threshold of 0.5.

# G   Datasheet

We include a datasheet based on the framework set forward by Gebru et al. [22]. Some parts of the proposed datasheet are omitted since our datasets are synthetic.

## G.1   Motivation

**For what purpose was the dataset created?**   The dataset was created to test, develop, and improve machine-learning models for financial crime detection. In particular, the datasets focus on identifying money laundering transactions.

**Who created the dataset and on behalf of which entity?**   Erik Altman on behalf of IBM.

**Who funded the creation of the dataset?**   IBM

## G.2   Composition

**What do the instances that comprise the dataset represent?**   The datasets are synthetic financial transaction networks. Each node in the network represents an account/entity and each directed edge represents a transaction from one account to another. Edge features detail the amount, currency, and type of transaction amongst other properties. The datasets also contain transactions that are labeled as money laundering. All the information is simulated. No real account or transaction details were used to create the datasets.

**How many instances are there in total?**   There are 6 datasets. Each dataset consists of one graph. The number of transactions (samples) ranges from 5M to 180M.

**Does the dataset contain all possible instances or is it a sample (not necessarily random) of instances from a larger set?**   The datasets are synthetic. Any number of transactions could be generated.

**What data does each instance consist of?**   When used for transaction classification, the edges can be considered instances. Each instance consists of a set of transaction features (incl., amount, currency, date, time, and type). In addition, each transaction is part of a whole network of transactions, and since the network topology plays an important role, the position of the instance in the whole network could be considered a "part" of the instance.

**Is there a label or target associated with each instance?**   Yes.

**Is any information missing from individual instances?**   No.

**Are there recommended data splits (e.g., training, development/validation, testing)?**   Yes.

**Are there any errors, sources of noise, or redundancies in the dataset?**   Not to the knowledge of the authors.

**Is the dataset self-contained, or does it link to or otherwise rely on external resources (e.g., websites, tweets, other datasets)?**   The dataset is self-contained.

**Does the dataset contain data that might be considered confidential (e.g., data that is protected by legal privilege or by doctor–patient confidentiality, data that includes the content of individuals' non-public communications)?**   No.

**Does the dataset contain data that, if viewed directly, might be offensive, insulting, threatening, or might otherwise cause anxiety?**   No.

## G.3 Collection Process, Uses, Distribution and Maintenance

Please refer to Section 4 and Appendix A for details about the generation process. The current usage is detailed in the paper and potential uses are described in Appendix B. The Kaggle page acts as the single source of distribution [50]. The dataset will be maintained there.

