# OpenReview forum: "Realistic Synthetic Financial Transactions for Anti-Money Laundering Models"
_NeurIPS.cc/2023/Track/Datasets_and_Benchmarks — NeurIPS 2023 Datasets and Benchmarks Poster_

### Official Review · Reviewer_w9iG · 2023-07-09
**The authors present an anti-money laundering (AML) dataset generator, which is calibrated to generate synthetic datasets with transactions that match real transactions as closely as possible. While AML is a serious real-world problem that should be addressed, I think this work has considerable ethical implications, because it might give rise to (or even incentivize) new Money Laundering schemes.**

**Rating:** 4
**Confidence:** 5
**Correctness:** The claims made in the submission are…
**Clarity:** The paper is well-written and clearly…

**Strengths:**

1. The addressed problem is a serious real-world problem.
2. The paper is well-written and the work well presented.
3. The transaction generation process is clearly described and is quite realistic.
4. The evaluation is well conducted.

**Additional Feedback:**

--

**Documentation:**

Six IT-AML datasets are now publicly available on Kaggle, which from an ethical perspective raises many questions concerning the incentivization of new Money Laundering strategies.

**Ethics:**

I think this work has considerable ethical implications, because it might give rise to (or even incentivize) new Money Laundering schemes.

**Limitations:**

See answers above.


**Opportunities For Improvement:**

1. I think this work has considerable ethical implications because it might give rise to (or even incentivize) new Money Laundering schemes.
2. While there is plenty of detailed expert knowledge incorporated in the generation scheme, the research aspects of this work are rather limited and weak (i.e., which research questions can I investigate by using the generated synthetic datasets?).

**Relation To Prior Work:**

Yes, it is clearly discussed how this work differs from previous contributions.

**Summary And Contributions:**

The authors describe the synthetic AML data generator in detail and demonstrate how the AML datasets generated can help to train Graph Neural Networks to detect money laundering activities. The generator is calibrated to generate synthetic datasets with transactions that match real transactions as closely as possible.

---

### Official Review · Reviewer_jAyB · 2023-07-09
**Very Interesting Work**

**Rating:** 6
**Confidence:** 3
**Correctness:** No issues in this aspect.
**Clarity:** The paper is informative and well-wri…

**Strengths:**

The contribution of the proposed IT-AML generator is significant as it utilizes a more advanced algorithmic design compared to existing methods like AMLSim. This improvement has the potential to establish a new standard for similar tools in the field of Anti-Money Laundering.

The quality of the research appears sound, considering the introduction of these novel tools and their potential for effective application.

In terms of relevance, the paper aligns well with the scope of the Datasets and Benchmarks Track, focusing on the development of synthetic datasets for Anti-Money Laundering purposes.

The paper demonstrates clarity in its writing style and structure, contributing to its overall cohesiveness and understandability.

**Additional Feedback:**

Detailed feedback comments can be seen in other sections.

**Documentation:**

Good

**Ethics:**

No ethics issues/concerns.

**Limitations:**

See my comments in "Opportunities For Improvement"

**Opportunities For Improvement:**

An important concern regarding the proposed agent-based simulator is the assurance provided by the authors regarding the reliability of the shared dataset as a benchmark. Considering the nature of agent-based simulations, even minor alterations in the settings could potentially result in significant variations in the generated data points. To address this concern, it would be valuable for the authors to engage in a more comprehensive discussion on the dataset's reliability.

Specifically, the authors should provide detailed insights into how stability and consistency in the generated data are preserved. This could involve elaborating on the validation procedures implemented to ensure the reliability of the generated dataset. Additionally, discussing any sensitivity analyses or robustness checks conducted to evaluate the dataset's consistency and its sensitivity to parameter variations would be beneficial. By providing a more thorough examination of these aspects, the authors can strengthen the confidence in the reliability and usefulness of the shared dataset as a benchmark for further research.

**Relation To Prior Work:**

The authors have adequately discussed how their work relates and differs from previous studies.

**Summary And Contributions:**

The paper introduces a synthetic financial transaction dataset generator along with a collection of synthetically generated Anti-Money Laundering (AML) datasets, aiming to address the challenges associated with money laundering. This approach provides an alternative to real data, which is often limited due to privacy and legal restrictions. Notably, the study presents an improved methodology by incorporating more sophisticated settings to simulate agent interactions for data generation, surpassing existing agent-based approaches. Additionally, the provided datasets serve as valuable benchmarks for model comparison and furthering research in the field of AML.

---

### Official Review · Reviewer_iXyM · 2023-07-21
**A new dataset generation method of Anti-Money Laundering.**

**Rating:** 7
**Confidence:** 3
**Clarity:** The paper is clear and well-written.

**Strengths:**

1. Novel Contribution: The paper presents a valuable contribution in the form of a realistic synthetic financial transaction dataset generator and AML datasets. This fills the gap of limited real-world data for training AML models and provides a standardized benchmark for comparisons.
2. Realism: The agent-based generator is calibrated to closely match real transactions, making the synthetic data more realistic and representative of actual money laundering scenarios.
3. Complete Ground Truth Labels: Unlike real data, the synthetic datasets have perfect tagging for laundering activity, enabling better training and evaluation of AML models.
4. Public Availability: The publicly available datasets on Kaggle facilitate research and encourage collaboration among researchers and practitioners working on AML detection.

**Additional Feedback:**

Referring to the above.

**Correctness:**

The claims made in the submission are correct. The dataset is constructed in a sound way. In the benchmark part, the evaluation methods and experiment design are appropriate and performed correctly.

**Documentation:**

There are sufficient details on data collection and organization. Sections 3.3 and 3.4 describes the IT-AML generator and the virtual world model. There provides deatils of how the synthetic dataset is generated.

**Ethics:**

As the research involves money laundering activities, ethical considerations should be addressed in terms of the potential misuse of synthetic data for illegal purposes. For example, if the model is used by money launderers to train better and more powerful models, money laundering will be more difficult to detect.

**Limitations:**

Refer to Ethics and other sections.

**Opportunities For Improvement:**

1. The paper highlights the advantages of using synthetic data, but it should also address potential discrepancies between model performance on synthetic data versus real data to ensure the practical applicability of the proposed approach.
2. The paper briefly mentions baseline scores for GNN-based models but lacks a comprehensive evaluation of various metrics for comparing different AML models' performances, which could strengthen the validity of the proposed approach. Moreover, the GNN baselines used in the paper are quite basic. More SOTA methods should be included.

**Relation To Prior Work:**

Table 1 compares the proposed model with previous models, which makes it clear that they have correspoding shortcomings. IT-AML differs from them. First, ITAML uses a virtual world model to mimic what happens in the real world. Second, it contains rich label information.

**Summary And Contributions:**

The paper introduces a synthetic financial transaction dataset generator and Anti-Money Laundering (AML) datasets. These datasets offer a realistic benchmark for training machine learning models to detect money laundering, overcoming the scarcity of real-world data. The proposed IT-AML approach provides complete ground truth labels for laundering activity, unlike real data. Experiments with Graph Neural Networks (GNNs) demonstrate their effectiveness in identifying laundering transactions. The publicly available datasets facilitate the development of AML models and enable the detection of previously unseen laundering patterns. Further research is suggested to address complex laundering patterns in datasets with low illicit ratios and to explore differentially-private graph topology and model sharing techniques.

---

### Decision · Program_Chairs · 2023-09-22

**Decision:**

Accept (Poster)

**Comment:**

In this paper, the authors have introduced  a synthetic financial transaction dataset generator and Anti-Money Laundering (AML) datasets. These datasets offer a realistic benchmark for training machine learning models to detect money laundering. Based on the discussion and current reviews, I recommend accept.